# Mean Shift Cluster Recognition Method Implementation in the Nested Sampling Algorithm

**DOI:** 10.3390/e22020185

**Published:** 2020-02-06

**Authors:** Martino Trassinelli, Pierre Ciccodicola

**Affiliations:** Institut des NanoSciences de Paris, CNRS, Sorbonne Université, 4 Place Jussieu, 75005 Paris, France; ciccopierre@gmail.com

**Keywords:** nested sampling, cluster analysis, mean shift method, Bayesian evidence, model comparison

## Abstract

Nested sampling is an efficient algorithm for the calculation of the Bayesian evidence and posterior parameter probability distributions. It is based on the step-by-step exploration of the parameter space by Monte Carlo sampling with a series of values sets called live points that evolve towards the region of interest, i.e., where the likelihood function is maximal. In presence of several local likelihood maxima, the algorithm converges with difficulty. Some systematic errors can also be introduced by unexplored parameter volume regions. In order to avoid this, different methods are proposed in the literature for an efficient search of new live points, even in presence of local maxima. Here we present a new solution based on the mean shift cluster recognition method implemented in a random walk search algorithm. The clustering recognition is integrated within the Bayesian analysis program NestedFit. It is tested with the analysis of some difficult cases. Compared to the analysis results without cluster recognition, the computation time is considerably reduced. At the same time, the entire parameter space is efficiently explored, which translates into a smaller uncertainty of the extracted value of the Bayesian evidence.

## 1. Introduction

At present, Bayesian methods are routinely used in many fields: astrophysics and cosmology [1,2,3,4,5,6,7,8], particle physics [9], plasma physics [10,11], machine learning [12] and many others [13,14]. In the past few years, they were also applied to nuclear [15,16] and atomic physics [17,18,19,20,21]. On one hand, one of the reasons for this success is related to the possibility of assigning a probability value to models (hypotheses) from the analysis of the same set of data in a very defined framework. In opposite to this, classical statistical tests and criteria (e.g., chi-square and likelihood ratio, Aikaike information criterion [22], etc.) are completely powerless if any defined preference does not emerge. On the other hand, the implementation of Bayesian methods is only now widely possible thanks to the recent relatively cheap cost of computation power. A large computing capability is in fact required for the fine exploration of the probability distribution of the model parameters. Unlike standard methods, which are mostly reduced to minimization/maximization problems (of the likelihood function or chi-squares), Bayesian approaches have to deal with non-trivial integrations in multi-dimensional space. One of the key points of Bayesian model selection is in fact the calculation of the *Bayesian evidence*, also called *marginal likelihood*, defined by
(1)$$E\left(M\right)\equiv P\left(Data|M,I\right)=\int P\left(Data|a,M,I\right)P\left(a|M,I\right)d^{J}a=\int L^{M}\left(a\right)P\left(a|M,I\right)d^{J}a.$$
It consists in the integral of the likelihood function $L^{M}\left(a\right)=P\left(Data|a,M,I\right)$ in the *J*-dimensional parameter space (with *J* the number of the parameters) weighted by the prior probability P(a|M,I) of the parameters a of a defined model M and where *I* represents the background available information. From the evidence, the probability of the model P(M|Data,I) is simply evaluated by the formula
(2)P(M|Data,I)∝E(M)P(M|I),
where P(M|I) is the prior probability of the model itself. The challenging part resides in the multi-dimensional integration of Equation (Equation 1). For this matter, different approaches have been developed in the past, some of them are Markov Chain Monte Carlo (MCMC) based techniques (see e.g., [14,23]) for the integration of $L^{M}\left(a\right)P\left(a|M,I\right)$. As an alternative, the *nested sampling* method has been proposed by Skilling in 2004 [24,25,26]. With this method, the multi-dimensional integral in Equation (Equation 1) is reduced to a one-dimensional integral and calculated. Because of its high-efficiency and relatively moderate calculation power requirement compared to other approaches, the nested sampling method is actually implemented in several data analysis codes such *Multinest* [3,27], *Diamonds* [28], *Polycord* [29], *UltraNest*, *DNest4* [30] and *Dynesty* [31] for the computation of the Bayesian evidence and posterior probability distributions. Because of its efficient sampling, nested sampling is also routinely used to study thermodynamic partition functions [32,33,34,35] and to explore potential energy landscapes of atomistic systems [36,37,38].

When several maxima of the likelihood function are present, nested sampling algorithm can however encounter problems with converging correctly. The parameter space exploration can become inefficient or exclude entire regions, which introduces systematic errors in the estimation of the evidence. In order to avoid such a problem, several solutions are proposed in the literature. Here we present an original approach based on cluster recognition with the *mean shift method*, one of the classic clustering algorithm widely used and included in the major machine learning libraries. This method is implemented in the program *NestedFit*, a code developed by one of the authors and described in details in [39,40].

An introduction to nested sampling and NestedFit code is presented in Section 2. The description of the mean shift algorithm, its implementation on NestedFit and the results of some tests are presented in Section 3. The article will end with a conclusive section (Section 4).

## 2. Nested Sampling and NestedFit

### 2.1. The Nested Sampling Algorithm

Nested sampling is based on the reduction of the multi-dimensional integral in Equation (Equation 1) for the evidence computation into a one-dimensional integral
(3)$$E\left(M\right)=\int_{0}^{1}L\left(X\right)dX.$$
*X* represents the normalized value of the volume, weighted by the prior probability P(a|I), of the portion of *J*-dimensional space of parameters where L(a) is higher than a certain value L:(4)$$X\left(L\right)=\int_{L(a)>L}P\left(a|I\right)d^{J}a.$$
Equation (Equation 3) is numerically calculated using the rectangle integration method subdividing the [0,1] interval in M+1 segments with an ensemble $\left\{X_{m}\right\}$ of *M* ordered points $0<X_{M}<\ldots <X_{2}<X_{1}<X_{0}=1$:(5)$$E\left(M\right)\approx \sum_{m}L_{m}\Delta X_{m},$$
where $L_{m}=L\left(X_{m}\right)$ isgiven by the invertible relation in Equation (Equation 4) and $\Delta X_{m}$ is simply given by $X_{m}-X_{m+1}$ or by the more accurate trapezoid rule $\Delta X_{m}=1/2\left(X_{m-1}-X_{m+1}\right)$. Each $\Delta X_{m}$ represents a slice of parameter space of nested hypervolumes defined by Equation (Equation 4), giving the algorithm its name.

The evaluation of $L_{m}$ is obtained by a recursive step-by-step exploration of the likelihood function by a Monte Carlo sampling. A collection of *K* parameter values $\left\{a_{k}\right\}$, called *live points*, corresponds to *K* random points $\left\{\xi_{1,k}\right\}$ in [0,1] interval. When the live point ${\tilde{a}}_{1}=a_{1,k^{\prime }}$ corresponding to the highest value of $\left\{\xi_{1,k}\right\},\xi_{1,k^{\prime }}=\max\left\{\xi_{1,k}\right\}$ (with $L_{1}=\min\left\{L\left(\xi_{1,k}\right)\right\}=L\left(\xi_{1,k^{\prime }}\right)\equiv L\left({\tilde{a}}_{1}\right)$ from Equation (Equation 4)) is discarded, the mean value of the interval occupied by the remaining $\xi_{k}$ points shrinks to
(6)$$X_{m}=\max_{k\neq k^{\prime }}\left\{\xi_{k}\right\}\approx {\left(\frac{K}{K+1}\right)}^{m}\approx e^{-m/K}$$
with, at this first step, m=1.

If a new live point $a_{new}$ is found with the condition $L\left(a_{new}\right)>L_{m=1}$, a new set of $\xi_{m=2,k}$ points is constructed and the next procedure iteration step starts. For each step, the discarded values ${\tilde{a}}_{m}=a_{m,k^{\prime }}$ are stored together with their corresponding likelihood values $L_{m}=L\left({\tilde{a}}_{m}\right)$. The $X_{m}$ are obtained by their average expectation value from Equation (Equation 6). Step by step, the nested volumes built with the condition $L\left(a\right)>L_{m}$ converge around the parameter space regions corresponding to high values of the likelihood function. When the algorithm converges, the evidence is evaluated from the different values $L_{m},\Delta X_{m}$ using Equation (Equation 5). From the set of collected values of the discarded live points ${\tilde{a}}_{m}$ and the associated weights $w_{m}=L_{m}\Delta X_{m}$, the posterior probability P(a|Data,M,I) can be determined. More details on the nested sampling algorithm and its implementation can be found in Refs. [3,24,25,26,27,41,42].

### 2.2. Bottleneck of Nested Sampling and Proposed Solutions

The difficulty of this elegant method is to efficiently find a new live point at each step within the hypervolume contour defined by $L\left(a\right)>L_{m}$. Codes that use the nested sampling method generally encounter difficulties to find new live points $a_{new}$ when several maxima of the likelihood function are present. In this case, the exploration of the parameter space becomes generally inefficient or can consider only one local maximum while introducing systematic errors in the estimation of the evidence. In order to avoid these problems, different strategies have been proposed in the literature. These strategies can be divided into two categories: with a cluster recognition algorithms and without cluster recognition, but with other improvements of the search algorithm for new live points.

A first attempt to improve the search of new live points for multimodal problems via MCMC has been proposed by Veitch and collaborators in 2010 [42]. Here 10% of the steps of the random walk are determined by a combination of three past points and not only the previous point of the Markov chain. In this way, a more efficient sampling is obtained without need of cluster recognition.

Another improved random walk method for nested sampling algorithm is the *diffusive nested sampling*, developed by Brewer et al. in 2011 [43] and implemented in *DNest4* [30] program. Here, the passage between maxima is facilitated by blurring the condition $L\left(a_{i}\right)>L_{m}$ for the parameter values explored by the MCMC, allowing to momentarily pass in regions with lower values of the likelihood function.

Alternatively to random walks, the use of single- or multi-particle trajectories have been implemented for improving the search of new points in complex landscapes of the function to maximize or minimize. This is the principle of Galilean and Hamiltonian Monte Carlo exploration [34,44]. In the first case, linear trajectories and reflection from hard boundaries, given by the minimal likelihood threshold value, are considered. In the second case, more complex trajectories are computed from the motion determined by the Hamiltonian function, like in molecular dynamics, assimilated here to the likelihood function.

In the case of the presence of several maxima, these methods significantly improves the search of new points but do not allow to pass from one maximal region to another, which limits their efficiency. A completely different approach has been proposed by Martiniani and collaborators in 2014 [45]. To take into account the presence of several maxima without recurring to cluster recognition, they suggest global optimization techniques to use the knowledge of identified local maxima and their statistical weight and then to perform parallel nested sampling in correspondence of each significant region.

A first solution with the use of a cluster recognition algorithm has been implemented in *Multinest* code already in 2008 [3,27]. Here, new live points are randomly selected in an ellipsoid that is defined by the covariance matrix of the present live points. Cluster analysis is used for partitioning of the parameter spaces in a series of ellipsoids. This is obtained by implementing the *k*-means clustering algorithm, which is triggered when the estimated volume occupied by the live points is much smaller than the ellipsoid volume estimated from their covariance matrix. A partition in two cluster is initially performed (k=2) and recursively repeated (always with k=2) to obtain an efficient partition of the space with many ellipsoids.

In the more recent *Polycord* program [29], where the search of new live points is based on the slice sampling (a MCMC that uses the live point covariant matrix to provide a probability distribution for the choice of the random walk direction), the cluster recognition is obtained by the *k*-nearest neighbor algorithm. Once the different cluster are identified, for each of them a parallel exploration and analysis via slice sampling MCMC is independently performed.

In the recent and very complete nested sampling code *Dynesty* [31], different sampling methods are proposed: from random uniform selections in ellipsoids, like Multinest, to a series of MCMC (random walks, slice sampling, …). Difficult cases with several likelihood maxima are treated by decomposing the parameter space in several ellipsoids via a cluster analysis (using *k*-means algorithm like Multinest), or spheres or cubes (with same radius/side, one per each live point implementing the *RadFriends* algorithm [46]) with no need of any cluster recognition technique.

In the following sections, we present a new alternative method based on an MCMC and where the mean shift algorithm is used for the identification of clusters. It is implemented in the existing nested sampling code *NestedFit*, which is briefly introduced as well in the next paragraph.

### 2.3. The NestedFit Program

NestedFit is a general-purpose code for the evaluation of Bayesian evidence and parameter probability distributions based on nested sampling algorithm. It is written in Fortran90 with some subroutines in Fortran77, and parallelized via OPEN- MPI. It is mainly developed and implemented for the analysis in the fields of atomic, nuclear and solid-state physics [16,39,40,47,48,49,50]. It is accompanied by a Python function library for visualization of the results and for automatization of series of analyses. In this publication we present the version 3.2 that has the cluster analysis of the live points as substantial upgrade with respect to older versions (see Ref. [39] for v. 0.7 and Ref. [40] for v. 2.2). In addition, in this new version some new improvements in the search of live point are also implemented. The source code is freely available in the repository https://github.com/martinit18/nested_fit.

The code requires two main input files: the main input file (nf_input.dat) where the analysis parameters are selected, and the data file, in the format (channel, counts) or (channel, y value, y uncertainty). Dependent on the data format, a Poisson or Gaussian statistics likelihood function is used. The function name in the input file indicates the model to be used for the calculation of the likelihood function. Several functions are already defined in the function library for different model of spectral lines. Additional functions can be easily defined by the user in a dedicated routine. Non-analytical or simulated profile models can be considered as well. In this case, one or more additional files have to be provided by the user for interpolation by B-splines using FITPACK routines [51].

Several data sets can be analyzed at the same time. This is particularly important for the correct study of physically correlated spectra at the same time, e.g., background and signal-plus-background spectra. This is implemented by using a global user-defined function composed by different models to describe each spectra but with common parameters between the models.

The main analysis results are summarized in one output file (nf_output_res.dat). Here the details of the computation, number of trials, number of iteration, can be found as well as the final evidence value and its uncertainty E±δE, the parameter values corresponding to the maximum of the likelihood function, but also the mean, the median, the standard deviation and the confidence intervals one, two and three sigma (68%, 95% and 99%) of the posterior probability distribution of each parameter. δE, or more precisely δ(lnE) is evaluated by running the nested sampling several time for different sets of starting live points. δ(lnE) is obtained by the standard deviation of the different values of lnE, the natural estimation to study the uncertainty of *E* [52,53]. The information gain H and the Bayesian complexity are also provided in the output. Data for plots and for further analyses are provided in separated files. The step-by-step information of the nested sampling exploration can be found in the largest output file that contains the live points used during the parameter space exploration ${\tilde{a}}_{m}$, their associated likelihood values $L_{m}$ and weight $w_{m}=L_{m}\Delta X$. From this file, the different parameter probability distributions and joint probabilities can be build from the marginalization of the unretained parameters.

Details of the NestedFit search algorithm are presented in the next section. Additional information can be found in Refs. [39,40].

### 2.4. NenstedFit Search Algorithm

The search of new live points in NestedFit is based on a random walk called *lawn mower robot* [39,40,54], which is represented in Figure 1a. It is composed by a sequence of *N* steps (or jumps, with *N* selected by the user) starting from a randomly chosen live point. Each step has an amplitude and direction given by the *J*-dimensional vector frσ where each component $fr_{j}\sigma_{j}$ is determined by factor *f*, selected by the user, a random number $r_{j}$ and the standard deviation of the current live points $\sigma_{j}$ with respect to the *j*th parameter. For an efficient covering of the entire parameter space, *f* and *N* should be chosen with the criterion
(7)f×N≳1
to explore regions within a distance of the order of one standard deviation around the starting point. Each new step, which correspond to a new parameter set $a_{n}$, is accepted if $L\left(a_{n}\right)>L_{m}$. If $L\left(a_{n}\right)<L_{m}$, a new set of $r_{j}$ is chosen. The number of total tries $n_{t}$ is recorded. The choice of the values for *f* and *N* is very critical and it could vary from case to case. *N* has to be large enough to lose the memory of the starting live point position, but an increase of it produces a linear increase in computation time. A similar situation arises for *f*. If it is too small, a strong correlation between live points is artificially created. If it is too large, many failures can occur. From our experience, a reasonable range of values is N=20−40 and f=0.1−0.2. In any case we suggest a visual check of the explored live points for detecting possible correlations.

If the number of failures becomes too high ($n_{t}\gg N$), two different strategies are implemented for finding a new live point. In the first one, schematically represented in Figure 1b, a new parameter set is determined by randomly choosing a point between the last failing chain point $a_{n}$ with $L\left(a_{n}\right)<L_{m}$ and the barycenter of the current live points. As for the *lawn mower robot* method, also this algorithm is due to Simons [54] but it was not implemented in the past versions of NestedFit.. The second method, represented in Figure 1c, consists of building a new synthetic live point $a_{new}$ from the jth components from distinct live points: ${\left(a_{new}\right)}_{j}={\left(a_{m,k}\right)}_{j}$ where *k* is randomly chosen between 1 and *K* (the total number of live points) for each *j*. If $a_{new}$$L\left(a_{new}\right)>L_{m}$, the new point is considered, otherwise another random live point is chosen as start of the random walk.

The two strategies are chosen randomly when $n_{t}=N_{t}$ ($N_{t}$ chosen by the user in the configuration file) and $n_{t}$ is reset to zero. As suggested by the schemes in Figure 1, the first one favors a re-centering of the live points. In the opposite, the second can more easily explore peripheral regions. This second strategy was in fact the only present in the previous versions of NestedFit (where also $N_{t}$ was a fixed parameter of the code), and it was developed to improve the search algorithm for multimodal cases facilitating jumps between maximal regions of the likelihood function [39,40].

If the entire above procedure is repeated subsequently too many times ($N\;N_{t}$, selected by the user), the cluster analysis, described in the following sections, is triggered for improving the search of new live points.

## 3. Mean Shift Clustering Algorithm and Its Implementation

### 3.1. Preliminary Tests and Considerations on Other Cluster Recognition Algoritms

Before the implementation of one particular cluster recognition method in NestedFit, different algorithms from classical machine learning libraries (https://scikit-learn.org as example) have been considered and some of them have been tested with simple Python scripts. For this purpose, we used different ensembles of live points issued from NestedFit runs on real data when convergence problems were encountered. We excluded a priori Density-Based Spatial Clustering of Applications with Noise (DBSCAN) method. This method is well adapted for detecting cluster with singular shapes (e.g., arc of a circle) without necessarily improving the implemented random walk algorithm that is based on the standard deviation of the recognized cluster. We then tested the Gaussian mixture method with the determination of the number of clusters based on the expectation-maximization algorithm. The results were not convincing and required external criteria for determining the number of clusters. For similar reasons, we excluded the *k*-means method that requires a preliminary choice of number of clusters and the *x*-means method that uses external criteria to determinate the best choice of *k*. We did not consider the recursive use of *k*-means with k=2, like in the *Multinest* code, to keep a simple cluster recognition implementation.

From these preliminary tests and considerations, the *mean shift* clustering algorithm [55,56] emerged for its simplicity of implementation, its robustness and, more importantly, because it does not require a choice of the number of clusters before the analysis.

### 3.2. The Mean Shift Algorithm for Cluster Recognition

Mean shift is a recursive algorithm based on the iterative calculation of the mean of points within a given region. Considering an ensemble $\left\{x_{i}\right\}$, for each point the mean value $m_{i}$ of the neighbor points $N\;H(x_{i})$ is calculated recursively via a kernel function $K(x_{i},x_{j})$ via
(8)$$m_{s,i}=\frac{\sum_{x_{s,j}\in N\;H\left(x_{s,i}\right)}K\left(x_{s,i},x_{s,j}\right)x_{s,j}}{\sum_{x_{s,j}\in N\;H\left(x_{s,i}\right)}K\left(x_{s,i},x_{s,j}\right)},$$
with s=1 and $x_{s=1,i}=x_{i}$ for the first step. Then the procedure is repeated considering instead of the initial points $x_{i}$, the mean values of the previous step, $x_{s,i}=m_{s-1,i}$, until convergence or the maximum number of allowed steps is reached. Different points belonging to the same cluster are identified by the vicinity of the final $m_{s,i}$ values.

With the present implementation, via a Fortran module in NestedFit, the identification of the neighbor points NH is determined by the Euclidean distance $d(x_{i},x_{j})<D$, with *D* selected by the user. Two choices of *K* are available: a flat kernel $K(x_{i},x_{j})=1$, and a Gaussian kernel $K\left(x_{i},x_{j}\right)=\exp\left(-d\left(x_{i},x_{j}\right)/\ell \right)$, with *ℓ* the bandwidth selected by the user. Before the implementation of the mean shift algorithm, the live points are normalized to their minima and maxima ${\left(x_{k}\right)}_{j}=\left[{\left(a_{m,k}\right)}_{j}-\min\left\{{\left(a_{m,k}\right)}_{j}\right\}\right]/\left[\max\left\{{\left(a_{m,k}\right)}_{j}\right\}-\min\left\{{\left(a_{m,k}\right)}_{j}\right\}\right]$ to have parameter *D* and *ℓ* dimensionless and in a fixed possible range [0,1].

At the end of the analysis, each live point has an additional flag indicating its belonging cluster that is used in the main NestedFit search algorithm.

### 3.3. Mean Shift Implementation in NestedFit

As written above, the cluster analysis is triggered when there are too many tries in the main search algorithm ($n_{t}=N_{t}\times N\;N_{t}$). Once the cluster analysis is performed, the algorithm restarts from a random live point but, instead of the standard deviation of whole ensemble of live points σ, only the standard deviation of the belonging *c*th cluster $\sigma_{c}^{cluster}$ is used for the random walk. Even if the cluster analysis is not perfect (e.g., too many or too few clusters are recognized), the generally smaller values of $\sigma_{c}^{cluster}$ compared to σ significantly improves the efficiency of the nested sampling. When the algorithm becomes inefficient ($n_{t}$ reach $N_{t}$), a new starting live point is chosen. When $n_{t}$ is becoming too high again ($n_{t}=N_{t}\times N\;N_{t}$), a new cluster analysis is performed and the calculation continues until the end of the evidence calculation. Because of the random selection of the starting live point, small clusters have small probability to be chosen, and naturally disappear (or eventually grow) to the advantage (disadvantage) to clusters with higher (lower) likelihood values during the nested sampling.

To illustrate the cluster recognition at work in NestedFit, two practical examples are considered. In both cases, a Gaussian kernel has been used with a relatively large value of D=0.5−0.6 in order to avoid having too many isolated clusters and ℓ=0.1−0.2, which ensures a good convergence of the algorithm. The cluster analysis is triggered after few failures, $N\;N_{t}=2-3$, with a relatively low number of maximal tries $n_{t}$ ($N_{t}=100-200$) to change search strategy quite often when it becomes critical. With these criteria, the cluster analysis is triggered only about 2−10 times for one entire nested sampling computation.

The first example consists of the analysis of a high-resolution X-ray spectrum corresponding to the helium-like $1s2p{\;}^{3}\;P_{2}\to 1s2s{\;}^{3}\;S_{1}$ intrashell transition of uranium obtained by Bragg diffraction from a curved crystal [57]. For the analysis of the spectra, we assume the presence of four Gaussian peaks with the same width and a flat background. The second analysis is related to the measurement of the single decay of H-like ${}_{\;61}^{142}$Pm ions to the stable ${}_{\;60}^{142}$Nd bare nucleus via electron capture. Here, an exponential decay with a sinusoidal modulation is used as a model, considered parameters are the relative amplitude, pulsation and phase (see Ref. [16] for more details). Both data sets, presented in Figure 2, are characterized by low statistic and the presence of many local maxima of the likelihood function, which makes them therefore difficult to analyze. In the first case, the possible permutations of the position of different peaks correspond to different maxima of the likelihood (4! = 24 maxima for four peaks). In the second case, the multimodal behavior is caused by the different possible combinations of phase and pulsation values and corresponding beats.

To observe the evolution of the nested sampling algorithm with and without cluster analysis in the first case, we represent in Figure 3 the evolution of one of the model parameters ${\left({\tilde{a}}_{m}\right)}_{j}$ relative to the position of one of the four Gaussian peaks as function of the step number *m* for ten different choices (tries) of starting live points. Different colors correspond to different values of the step weight $w_{m}=L_{m}\Delta X_{m}$. Parameters with higher values or $w_{m}$ had a higher influence on the final evidence and probability distributions P(a|Data,M,I). When the cluster analysis was not implemented (Figure 3 (top)), each try slowly converged to one likelihood maximum only, which corresponds to one of the four possible positions. The convergence in different maxima produced as consequence a spread of the values of the Bayesian evidence *E*.

In contrast, when the cluster analysis was turned on (Figure 3 (bottom)), all four possible peak positions were considered at the same time and were equally explored for any try. The convergence improvement was directly observable in the smaller value of uncertainty of the evidence *E*. When the cluster analysis was off, we had lnE=−320.52±1.71 and lnE=−323.22±0.17 when it was on. These results were obtained with f=0.1 for the analysis without clusters and f=0.2 for the analysis with it, N=20 and K=2000 for both cases. The uncertainty of the previous values, and for all following evaluations, was obtained from the standard deviation of 16 different lnE values obtained running the analysis with 16 different sets of starting live points. The smaller value of *f* for the run without clusters was chosen to reduce the computation time, which was still about eight times longer than with the cluster analysis. It is interesting to note that, surprisingly, the two main values with and without cluster analysis werere compatible (note: a difference of 0.9 in the lnE corresponds to about two sigmas [58]), without a systematic shift due to the exploration of a smaller parameter space. Only the associated uncertainty significantly changes.

To better visualize the cluster analysis process, a 3D presentation of the evolution of three components of ${\tilde{a}}_{m}$, relative to the position of three peaks is presented in Figure 4. Each image is obtained just after a cluster analysis, where different clusters are represented by different colors. To note, the analysis was triggered only a few times (four times for this selected example with $K=2000,N_{t}=200,N\;N_{t}=2$ and with a Gaussian kernel with D=0.6 and ℓ=0.2), showing the efficiency of the clustering recognition in the search of new live points (for about 60000 steps for each run). After the first run of the mean shift analysis, only a large cluster (and few isolated live points) were identified. In the following cluster analysis, all 24 different maxima likelihood regions were correctly identified.

The correct and simultaneous identification of all maxima translated to a more regular histogram of probability distributions evaluated from the nested sampling outputs. This is shown in Figure 5, where the 2D histogram relative to the joint probability of position and amplitude of one of the peaks is presented for the analysis with and without cluster recognition. When the cluster analysis was not implemented, the presence of very localized maxima of the probability distribution reflected the pathological behavior of the nested sampling convergence to only one of the likelihood maxima. On the contrary, a much smoother distribution of the joint probability was present when the cluster analysis was on.

A more quantitative measurement on the cluster analysis was obtained by varying the number of used live points *K*. As it can be observed in Figure 6 (left, top), the final evidence did not change significantly with *K*. In opposite, the evaluated uncertainty (in blue) changed by several orders of magnitudes and was systematically larger than its theoretical estimation (in black) $\delta \left(\log E\right)\approx \sqrt{H/K}$ [25,52], where H is the information gain. When the evaluated evidence uncertainty was plotted in logarithmic scale (Figure 6 (left, bottom)), it can be observed that, for high values of *K* (≥500), δ(logE) was proportional to $1/\sqrt{H/K}$ as expected ($\delta \left(\log E\right)\propto K^{c}$ with c=−0.52±0.02), but was systematically higher by a factor of about 1.6 than the estimated accuracy (not shown in the bottom figure). When *K* was too low (K<500 in the present case), even with the cluster analysis, the nested sampling algorithm could not efficiently explore the 24 minima producing a systematic increases of δ(logE).

As expected, the computation time (equivalent for one single CPU) per set of live points grew almost linearly with *K*. A simple fit gives an exponential dependency $\propto K^{c}$ with c=1.13±0.01. A significant deviation was observed for K=10,000. In this case the cluster analysis, which number of operations was proportional to $K^{2}$, significantly contributed to the total computation time.

Cluster analyses of above results were obtained all with the same set of parameters: with a Gaussian kernel and with $D=0.6,\ell =0.2,N_{t}=200$ and $N\;N_{t}=2$. The exploration of the algorithm efficiency as dependence of these parameters is investigated and the corresponding results are resumed in Figure 7, where the final evidence values and required computation time are presented for different parameter sets. Several cases were considered with flat and Gaussian kernel, indicated in the label by ‘f’ and ‘g’, respectively, and different values of *D* and *ℓ*, indicated in the label as well (only *D* for the flat kernel). As it can be noticed, for too small values of *D* and *ℓ* the final accuracy was poor. This is related to the identification of too many and too small clusters that finally induced an inaccurate, but fast, exploration of the parameter space. On the opposite, for too large values, one or very few clusters were identified. In these cases, the cluster algorithm was called very often without really improving the situation but increasing significantly the total computation time. Gaussian kernel proves to be more robust and flexible than a flat kernel, probably due to the presence of the counter-reaction of the two parameters. The optimal parameter choice depended on the specific problem and the values of $N_{t}$ and $N\;N_{t}$. It was generally observed that low values of $N_{t}$ allowed for changing starting live point often enough improve the efficiency of the algorithm. $N\;N_{t}$ had to be adapted to trigger enough times the cluster analysis, but not too often.

The analysis of the other considered case was characterized by a completely different cluster evolution. In Figure 8 we represent the amplitude, pulsation and phase of the modulation values after each cluster analysis. After the first run, several clusters were identified even if no clear structures were visible. In the following analysis, a very complex landscape was drawn, with many clusters and with very narrow values in omega. Even if characterized by very different sizes for the different parameters (even after their normalization), different clusters were well identified by the mean shift algorithm.

The complex dependency on the modulation pulsation ω is also presented in Figure 9, where its evolution as function of the nested sampling step is represented for two different choices of starting live points. It can be observed that the rich landscape of the likelihood value as function of ω was well reproduced for each try, demonstrating the efficiency of the cluster analysis implementation once again.

As in the previous example, similar values of the Bayesian evidence were found: lnE=−1921.54±0.12 without cluster analysis and −1922.04±0.21 with. In contrast to the previous case, the uncertainty for the analysis without cluster analysis was very small. This was mainly caused by the choice of the value f=0.014 (and N=40, K=5000), a very small value compared to the value set for the analysis with cluster analysis (with f=0.1, N=20, K=5000). This small value of *f* contradicted in fact also the recommendation from Equation (Equation 7) with the risk to introduce some systematic errors in the computation. It was however required for insuring the convergence of the computation, which was otherwise impossible without cluster analysis. Like the previous example, the computation time without cluster analysis was in the best case about eight times longer than with the cluster analysis.

When the number of live points *K* was varied, keeping the other parameters fixed (Gaussian kernel with $D=0.6,\ell =0.2,N_{t}=100$ and $N\;N_{t}=3$), we could observe in Figure 6 (right) a similar tendency for the results as in the previous case. The estimated evidence accuracy was found to be proportional, as expected, to $1/\sqrt{K}$ ($\delta (\log E\propto K^{c}$ with c=−0.48±0.08). Here too, δ(lnE) was by a factor of 4.4–5.5 higher than the estimated accuracy. Because of the presence of less local minima than in the case of the four Gaussian peaks problem, the evaluated accuracy followed the proportionality to $1/\sqrt{K}$ down to K=100. An almost linear dependency of the computation time on *K* was visible in this case too (CPU time $\propto K^{c}$ with c=1.13±0.01), with a significant deviation for K=10,000 due to the high cluster analysis requirements for high *K*.

These two examples show the general behavior of the cluster algorithm and its dependency on the parameters choice. However, each case can be different and the user should vary the different parameters to reach the required accuracy. A general and simple suggestion is to use a large number of live points to efficiently explore the whole parameter space. This is crucial when multiple local maxima of the likelihood function are present to avoid missing one of them. This is an important requirement even when a cluster analysis is available.

## 4. Conclusions

We present a new application of cluster recognition to a nested sampling algorithm for the evaluation of the Bayesian evidence and posterior parameter probability distributions. For this matter, we selected the method of the mean shift, a robust and simple classical cluster recognition method widely used in the machine learning community. This clustering algorithm proved itself well adapted to critical data analysis when several likelihood maxima are present. It has been implemented in the program NestedFit and tested with two different benchmark cases, proving its efficiency in exploring the parameter space without excluding any region. As a consequence, the computation time is reduced by a factor at least eight. At the same time, a smaller value of the estimated evidence uncertainty is obtained. As a result from study the dependency on the different algorithm parameters, a sufficiently high number of live point should be always used, even when the cluster analysis is implemented, to efficiently explore all local likelihood maxima. Moreover for a good efficiency of the mean shift cluster recognition, its typical parametric distances (*D* and *ℓ*, the maximal neighbours distance and the bandwidth of the Gaussian kernel) should neither be too small or too large. In one case very low accuracy, but fast computation is obtained, in the other case the computation time increases too much.

In this article we explore only the implementation of the mean shift algorithm for cluster recognition. In the future, we plan to explore other methods like the *k*-nearest neighbours and the *x*-means method, successfully used in other nested sampling codes, and compare NestedFit performances with these codes in benchmark cases.

## Figures and Tables

**Figure 1 entropy-22-00185-f001:**
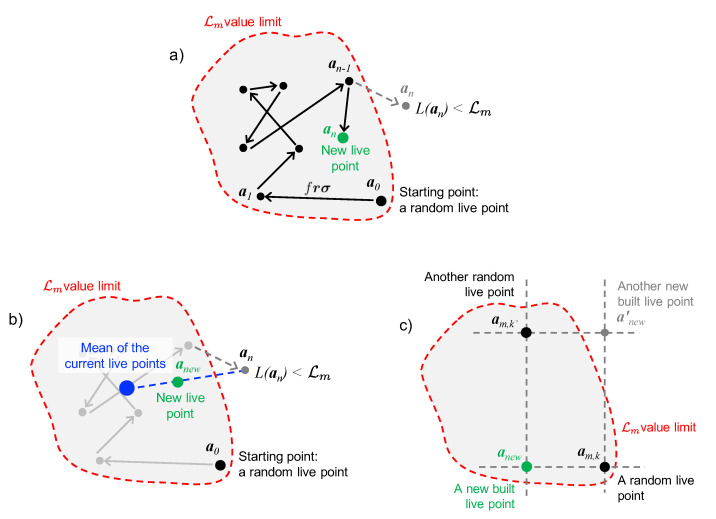
Graphical presentation of the different search algorithms discussed in the text. (**a**): Exploration of the parameter volume via the *lawn mower robot* for finding a new live point. (**b**): Search of a new live point from the parameter set $a_{n}$ outside the limit $L\left(a\right)>L_{m}$ and the barycenter of the current live points. (**c**): Construction of the new live point from different coordinates of the current live points.

**Figure 2 entropy-22-00185-f002:**
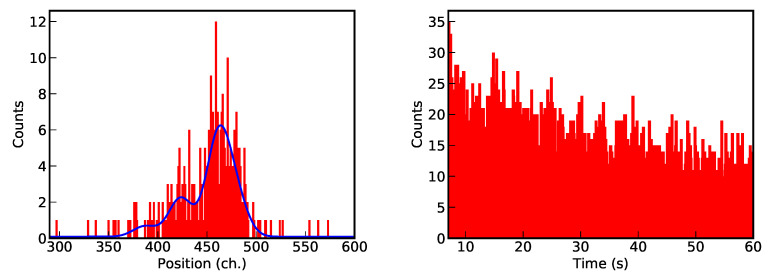
Data corresponding to the high-resolution X-ray spectrum of the helium-like uranium $1s2p{\;}^{3}\;P_{2}\to 1s2s{\;}^{3}\;S_{1}$ intrashell transition obtained by Bragg diffraction from a curved crystal [57] (**left**) and of the single decay of H-like ${}_{\;61}^{142}$Pm ions to the stable ${}_{\;60}^{142}$Nd bare nucleus via electron capture [16] (**right**).

**Figure 3 entropy-22-00185-f003:**
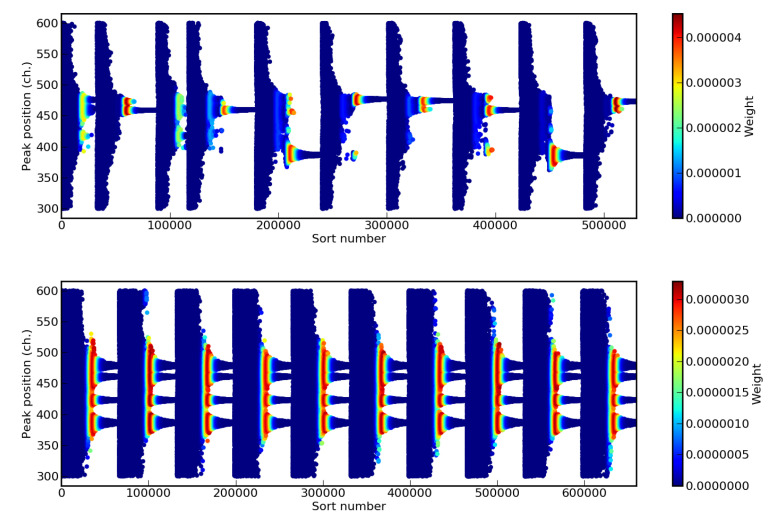
Evolution of one of the components of the discarded live points ${\tilde{a}}_{m}$ relative to the position of one of the four considered Gaussian peaks (see text) as function of the nested sampling step and for ten different choices starting live points. Results relative to the analysis without (**top**) and with the cluster analysis (**bottom**).

**Figure 4 entropy-22-00185-f004:**
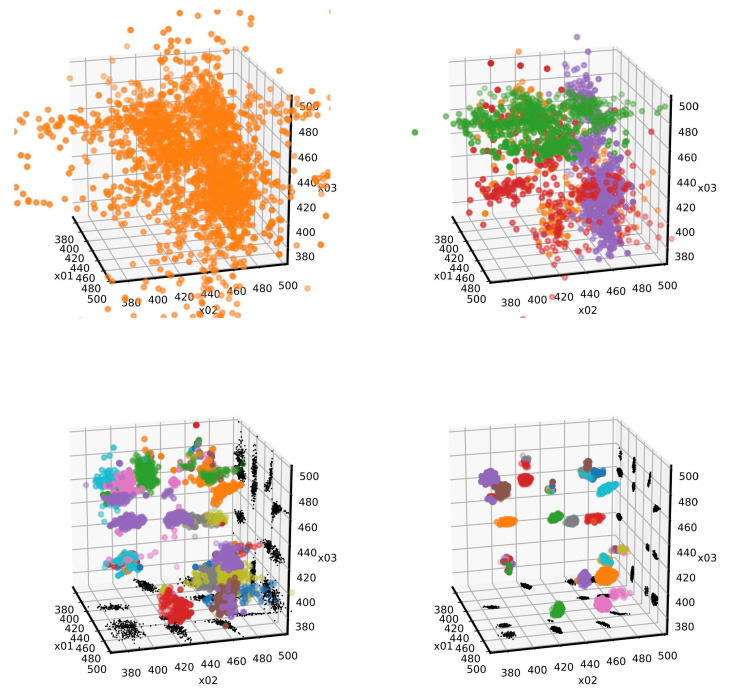
Results of the cluster recognition corresponding to the analysis of four Gaussian peaks. The position of three peaks are represented. Different colors represent different identified clusters. In black, the projection to some planes are represented. The 24 likelihood maxima (corresponding to the 4! permutation of the position of four peaks) are well visible.

**Figure 5 entropy-22-00185-f005:**
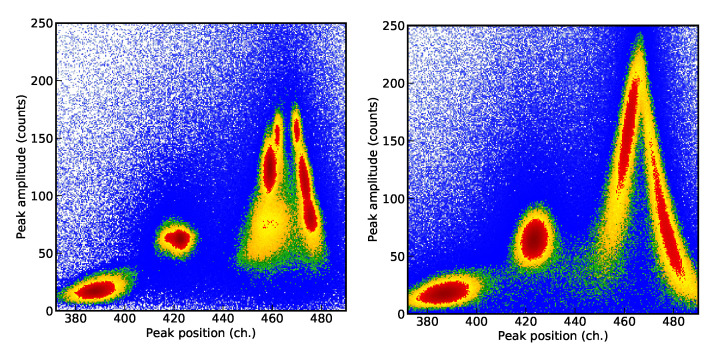
Joint probability distribution of the position and amplitude of one of the four considered peaks obtained without (**left**) and with cluster analysis (**right**).

**Figure 6 entropy-22-00185-f006:**
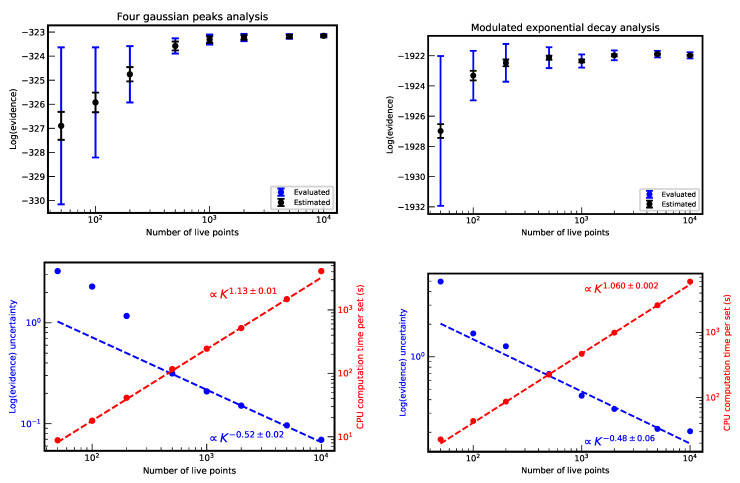
(**Top**) evaluation of the logarithm of the evidence for different number of live points *K* for the four gaussian peaks analysis (left) and the modulated exponential decay (right). In blue are indicated the uncertainty values evaluated by the results of 16 different run for each case. In black the theoretical uncertainty $\sqrt{H/K}$ estimated from the information gain H. (**Bottom**) dependency of the evaluated uncertainty and CPU time with *K*. The dashed lines are the fits with power laws, which results are also shown. Data relative to K<500 and K>5000 are excluded for the fit of logE uncertainty and CPU time, respectively.

**Figure 7 entropy-22-00185-f007:**
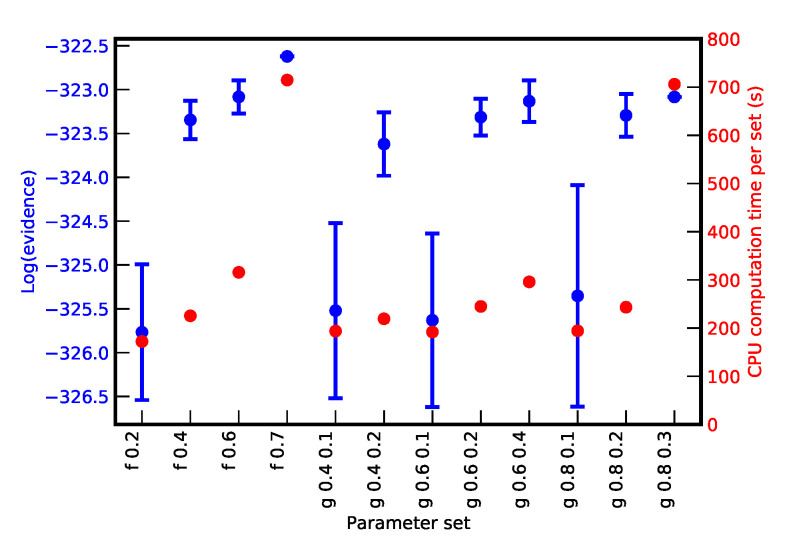
Values of logE and CPU time for different choices of parameter values of the cluster recognition algorithm. Uncertainties of the evidence relative to the labels ‘f 0.7’ and ‘g 0.8 0.3’ are not evaluated because of the large computation time corresponding to these cases.

**Figure 8 entropy-22-00185-f008:**
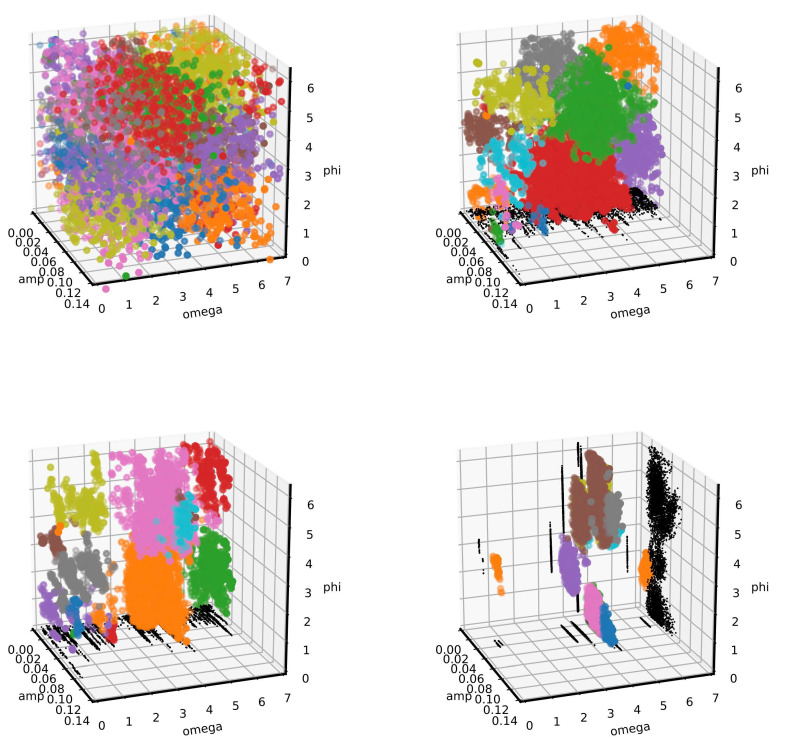
Results of the cluster recognition corresponding to the analysis of the modulation of the exponential decay. The relative amplitude, pulsation and phase are represented. Different colors represent different identified clusters. In black, the projection to some planes are represented.

**Figure 9 entropy-22-00185-f009:**
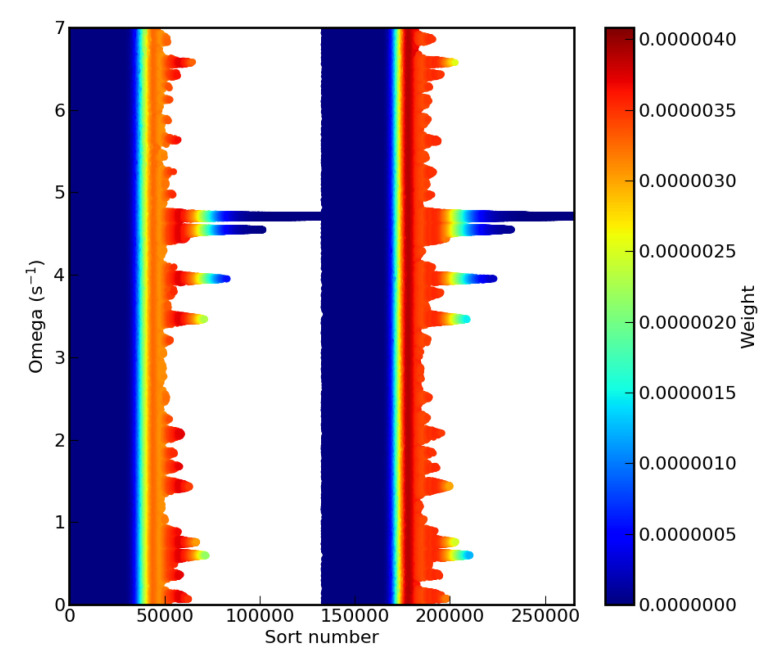
Evolution of one of the components of the discarded live points ${\tilde{a}}_{m}$ relative to the pulsation ω of the modulation of the single ion exponential decay (see text) as function of the nested sampling step with cluster analysis and for two different starting live points selections.

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
