# Peer review of "Mean Shift Cluster Recognition Method Implementation in the Nested Sampling Algorithm"

_entropy, 2020, doi:10.3390/e22020185_

Round 1
Reviewer 1 Report
The authors present their work on using the nested sampling algorithm, which is a useful and increasingly more popular algorithm for the calculation of the Bayesian evidence. Nested sampling has found applications in a wide range of problems and research areas, thus the proposed improvements to the practical implementation of the technique has a potentially wider impact, not just within the field of Bayesian statistics. I recommend the manuscript for publication after minor improvements.
It is not entirely clear from the manuscript whether the NestedFit code available for testing or using? Would it be possible to include a URL address pointing to e.g. a repository?
The authors demonstrate the new algorithm on different examples. It would be beneficial to know a bit more details on these calculations, how they chose the run parameters (e.g. the number of live points) and how this choice effects the outcome. Would it be possible to perform the tests on the same functions using a different number of walkers for example?
The authors might wish to mention other areas where the nested sampling method has been taken up, e.g. sampling the potential energy landscape of atomistic systems (Partay et al. J. Phys. Chem. B 2010, 114, 10502) and particularly different attempts to improve the exploration of the space in these cases of high-dimensional and highly multimodal functions. E.g. by utilising priory acquired data on the minima (maxima) regions (Martiniani et al. Phys. Rev. X 4, 031034 (2014)) and introducing a walk (analogous to molecular dynamics) which is more efficient in these cases than a simple random walk (Baldock et al. Phys. Rev. E 96, 043311 (2017)).
The manuscript is generally clear and reads well, but a thorough proofreading would be necessary to correct the several typos and small mistakes, and to improve the flow of the text. A couple of examples from pages 6 and 7
Page 6 line 142: “…at it was developed” as it was developed?
Page 6 line 157: “…instead of the of the initial…”
Page 6 line 158: “…belonging to a same cluster…” to the same cluster
Page 7 line 176: “continue” continues
Page 7 line 195: “that makes difficult the analysis” that makes the analysis difficult
Page 7 line 205-6: “randomly selected overt the four possible.” ???
Line 262 in Conclusions: “… by a factor of at least height” should be a “..at least eight”
Author Response
Please find in the attached file the detailed reply.

Reviewer 2 Report
This paper applies the 'mean shift' clustering algorithm to mode identification in nested sampling, and demonstrates that it is indeed effective in improving the accuracy of posterior sampling and evidence computation in the context of nested sampling.
The paper provides a good review of existing literature and a sufficient summary of background material. It demonstrates that the mean shift algorithm implemented in NestedFit improves the robustness of predictions, in comparison to when no clustering is implemented.
I judge the paper to be well-motivated, as the effect of clustering algorithm on nested sampling results has not been greatly explored in the literature. However, different clustering algorithms are something that designers of nested sampling algorithms in general explore offline before settling on their choice.
In order for me to recommend acceptance of this paper, the authors must provide a more thorough demonstration as to why or whether mean shift is preferable to other clustering algorithms (such as those they refer to in the first paragraph of 3.1) by explicit comparison of results. At the moment, the paper only demonstrates that clustering is important (which is known) rather than that mean shift clustering is preferable to other alternatives. It would also be appropriate to apply alternative nested samplers such as MultiNest, PolyChord and dynesty to the same examples so as to demonstrate whether mean shift is superior, or just equivalent to X-means, k-nearest neighbors or k-means. With these modifications, I would be happy to recommend for acceptance.
Author Response
Please find in the attached file the detail reply.

Round 2
Reviewer 1 Report
The authors implemented the suggestions made by the referees, and in my opinion the manuscript can be accepted for publication in its present form.
Reviewer 2 Report
The authors have not satisfactorily addressed the concerns raised in my previous review, on the grounds that it would take "longer than the standard time demanded for manuscript revision".
I cannot recommend acceptance of this manuscript until these additions/amendments are included, even if it takes longer than the standard time allowed.